# Predicting Injury Status in Adolescent Dancers Involved in Different Dance Styles: A Prospective Study

**DOI:** 10.3390/children7120297

**Published:** 2020-12-16

**Authors:** Damir Sekulic, Dasa Prus, Ante Zevrnja, Mia Peric, Petra Zaletel

**Affiliations:** 1Faculty of Kinesiology, University of Split, 21000 Split, Croatia; mia.peric@kifst.hr; 2Faculty of Sport, University of Ljubljana, 1000 Ljubljana, Slovenia; dasa.prus@fsp.uni-lj.si (D.P.); petra.zaletel@fsp.uni-lj.si (P.Z.); 3Clinical Hospital Split, 21000 Split, Croatia; antezevrnja17@gmail.com

**Keywords:** musculoskeletal injury, sports, exercise, risk factors, protective factors

## Abstract

The positive effects of dance on health indices in youth are widely recognized, but participation in dance is accompanied with a certain risk of injury. This prospective study aimed to investigate injury occurrence and to evaluate the possible influences of specific predictors on the occurrence of musculoskeletal problems and injuries in adolescent dancers. Participants were 126 dancers (21 males; 11–18 years), who were competitors in the urban dance, rock and roll, and standard/Latin dance genres. Predictors included sociodemographic factors, anthropometric/body build indices, sport (dance) factors, and dynamic balance. The outcome variable was injury status, and this was evaluated by the Oslo Sports Trauma Research Centre Overuse Injury Questionnaire (OSTRC). Predictors were evaluated at baseline, and outcomes were continuously monitored during the study period of 3 months. During the study course, 53% of dancers reported the occurrence of a musculoskeletal problem/injury, and dancers suffered from an average of 0.72 injuries over the study period (95% CI: 0.28–1.41), giving a yearly injury rate of 280%. Gender and dance styles were not significantly related to the occurrence of injury. Higher risk for injury was evidenced in older and more experienced dancers. Dynamic balance, as measured by the Star Excursion Balance Test (SEBT), was a significant protective factor of injury occurrence, irrespective of age/experience in dance. Knowing the simplicity and applicability of the SEBT, continuous monitoring of dynamic balance in adolescent dancers is encouraged. In order to prevent the occurrence of musculoskeletal problems/injuries in youth dancers, we suggest the incorporation of specific interventions aimed at improving dynamic balance.

## 1. Introduction

The importance of participating in sufficient physical activity (PA) in childhood and adolescence and the benefits of PA on physical health, mental health, academic performance, and social well-being have been well proven [1,2,3,4,5]. However, the decrease in participation in PA in adolescence is recognized as a global problem, which is additionally important since a decrease in PA may extend into adult life [6,7,8]. Despite the fact that Slovenian children and adolescents are among the most physically active in the world, their levels of PA have decreased in recent years, and this problem is especially exacerbated during adolescence [9].

To prevent a decrease in PA, various interventional programs have been constructed to encourage children and adolescents to be more physically active [10,11]. Since such programs should be motivational and attractive, dance is a perfect tool, as it fulfils not only PA requirements but also enables the development of social skills and expressiveness in youth [12,13,14]. As some individuals begin to dance very early in their childhood, dance is a particularly suitable form of PA that can be continuously applied whether in a professional or recreational context, throughout a participant’s lifetime [15,16,17].

However, as with any other PA, dance participation is accompanied by a certain risk of injury, even in youth [17,18,19]. Studies have reported that 20–84% of dancers have suffered from a musculoskeletal (MS) injury at least once in their career, and an even higher percentage (95%) have suffered from MS pain [18,20,21]. MS injuries vary by dance style, and by far the most common dancers to experience MS injuries are ballet dancers, followed by urban dancers (i.e., breakdance, hip-hop, locking, popping, house) and modern dancers [16,22,23].

Collectively, studies have confirmed higher injury rates in older dancers and/or shown a correlation between a dancer’s age and the occurrence of injury [24]. Further, due to characteristics of the dance activity and specifics of artistic expression, anthropometric/body build indices are known to be associated with injury risk, with a higher risk of injury in dancers with longer body segments and more body fat [25]. Studies have also confirmed that a there is a higher risk for injury occurrence in female dancers than in their male peers, which is connected to the most commonly injured body location in dancers (the knee), whereby the larger Q-angle in females potentially translates into greater forces of the quadriceps being applied to the patella and encourages mal-tracking [26,27]. Finally, a recent study provided evidence of a possible influence of dynamic balance as a protective factor against injury occurrence in dance, which was explained by the connection between dynamic balance and a dancer’s ability to sustain equilibrium by keeping their body over its base of support [17,28].

Irrespective of the well proven benefits of dance and the risks of participation in dance, only a few studies have exclusively examined youth dancers with regard to injury prevalence and factors of influence. In brief, US authors examined adolescent dancers (avg. age 15.3 y) at a liberal arts high school dance program over a one-year period (school year). Dancers self-reported 112 injuries (avg. 2.8 of self-reported injuries per dancer), and older age turned out to be the prevalent risk factor associated with self-reported injuries [24]. In a study of injury patterns in young, non-professional dancers, advanced age and increased exposure to dance (i.e., age) were also proven to be correlated with an increased prevalence of injury in girls (age 8–16 y) [29]. Meanwhile, there is an evident lack of studies that have prospectively examined factors associated with injury occurrence in youth dancers involved in different dance styles.

Therefore, the aim of this study was to prospectively analyze the injury occurrence in adolescent dancers involved in different dance styles (urban dance, standard/Latin dance, rock and roll) and to evaluate the possible factors influencing the occurrence of MS problems and injuries. Specifically, based on results of previous studies on different dance styles, we were particularly interested in anthropometric/body build indices, sociodemographic and dance factors, and dynamic balance as factors of possible influence on injury status in adolescent dancers. We hypothesized that the studied factors significantly influence the injury status of adolescent dancers.

## 2. Materials and Methods

### 2.1. Participants and Design

A total of 126 young dancers (21 males) involved in urban dance (breakdance, hip-hop, locking, popping, house; *n* = 99), rock and roll (*n* = 14), and standard and Latin-American dance (*n* = 13) all aged 11–18 years (mean age 15.66 ± 1.57) from Slovenia participated in the present study. Participants were selected on the basis of their status in dance sport, and all participants should be regular competitors at national and international levels under the auspices of the Slovenian Dance Association (SDA) and the International Dance Organization at the study baseline. The participants had an average of 7.8 ± 2.78 years of dance experience and trained for 7.09 ± 4.56 h per week at their respective dance schools. Based on (i) a previously evidenced injury occurrence of 10% [17], (ii) a population sample of 621 registered adolescent dancers in Slovenia for the studied year, (iii) margin of error of 0.05, and (iv) confidence level of 0.95, the required sample size for this investigation was calculated to be 114 participants (calculated by Statistica, Tibco Software Inc., Palo Alto, CA, USA)

In 2018, when the study was done, there were 23 dance teams/schools with registered adolescent dancers in Slovenia. All schools were invited to participate in the study by the SDA, and dancers were informed about the study’s aims, protocol, potential benefits, and risks. The participants were asked to provide consent from their parents, and participation was voluntary and anonymous. Dancers were qualified for inclusion if they were (i) minors (18 years or younger at the end of the study—see later for design), (ii) officially registered as a competition participant in the International Dance Organization, and (iii) participating in at least two dance trainings per week. Exclusion criteria included age of 18+ at the end of the study, non-regular participation in dance training (less than two dance trainings per week as defined by main coach), and injury/sickness during the baseline testing (see later for details on testing). Participants willing to be enrolled in the study were invited to screening at the Institute of Sport (University of Ljubljana, Faculty of Sport, Ljubljana, Slovenia). Involvement in the research was voluntary, and participants’ personal data were protected with an identification code, known only to the main/head researcher. The study protocol was approved by the Ethics Committee of the University of Ljubljana, Faculty of Sport, Ljubljana, Slovenia (Ref. number: 1175/2017). This prospective study included testing, which was done at baseline and during a follow-up period. Baseline testing of predictors (see later for details) was done in December 2018–January 2019, and follow-up testing was performed continuously over a period of 3 months after baseline and included an analysis of outcome.

### 2.2. Variables

Predictors included sociodemographic characteristics, anthropometric indices, dance factors, and dynamic balance (predictors). The outcome variable in this study was injury status.

#### 2.2.1. Sociodemographic and Dance Factors

Sociodemographic variables included age and gender. For the dance factors, dancers were asked about their (i) dance style (urban, standard/Latin (S/L), or rock and roll (RNR)), (ii) experience in dance (in years), (iii) age when they started to practice (later transformed into years of experience in dance), (iv) number of weekly training sessions, and (v) hours of weekly training.

#### 2.2.2. Anthropometric and Body-Built Indices

Anthropometrics included (i) body height (in 0.5 cm) and (ii) body mass (in 0.1 kg), both measured with standardized techniques and calibrated equipment and (iii) calculated body mass index (BMI; in kg/m^2^) and body composition indices (body fat percentage (BF%), and lean body mass (kg). Body composition was measured by bioelectrical impedance analysis with the InBody 720 Tetrapolar 8-Point Tactile Electrode System (Biospace Co. Ltd., Seoul, Korea) [30,31].

#### 2.2.3. Dynamic Balance

The Star Excursion Balance Test (SEBT), a functional screening tool, was used to measure balance performance in dancers. The test was designed to assess dynamic lower extremity balance, monitor rehabilitation progress, screen for deficits in dynamic postural control due to MS injuries, and identify athletes at high risk for lower limb injuries [32]. Performance of the test requires good balance, flexibility, strength, and coordination of the lower extremities. Although some authors have reported contradictory results regarding the accuracy of the SEBT test and its modifications as a predictor of an athlete’s risk of injury, the majority of recent studies have confirmed SEBT as one of the most prominent non-equipment screening tools to measure dynamic balance of the lower extremities [33]. SEBT has previously been shown to be a reliable measure and has been validated for use as a dynamic test for predicting the risk of lower limb injury [33]. Furthermore, the results of a recent systematic review showed that the SEBT has great inter- and intra-rater reliability [34]. The SEBT consists of eight-line grids, extending from the center point, with 45 degree angles between them. Every direction poses different demands and combinations regarding each motor ability in the frontal, sagittal, and transverse planes. Grids were taped on the floor with adhesive tape marked with centimeters. Individual verbal instructions and a demonstration were given to each participant by the same examiner, who then supervised the proper execution of the test. The participants took a unilateral position, with the stance foot in the center of the grid. Dancers had to reach down all of the marked lines as far as possible, using the non-stance leg, and then return with their reach leg back to the center of the grid, while maintaining a unilateral position. Dancers kept their hands flexed at the iliac crest throughout the test protocol. A result was not considered if (i) the dancer was not able to maintain the single-leg stance, (ii) the dancer changed the position of the foot during the trial (lifted their heel or toes off the floor, rotated the foot), (iii) the dancer’s weight was transferred onto the reaching foot, (iv) the dancer’s hands did not remain on their hips, or (v) the dancer was not able to firmly maintain the start and return position. The reaching distances were measured to centimeter accuracy and normalized to the % leg length of the participants. The variables observed in this investigation included the normalized SEBT performance when participants were standing on their right leg (R1–R8) and left leg (L1–L8). All dancers were evaluated by same examiner in the same facility (Faculty of Sport, Ljubljana, Slovenia). Measurement of the SEBT is presented in Figure 1.

#### 2.2.4. Injury Status

Injuries were recorded using the Oslo Sports Trauma Research Center Overuse Injury Questionnaire (OSTRC) [35]. Dancers responded to the OSTRC at baseline and prospectively once per week over the course of the study. At baseline, participants were personally asked about injury occurrence in the 3 month period before the testing. A digital form of the questionnaire was sent to participants by e-mail once a week. Additional individual reminders were sent to participants who did not provide any data for the preceding week. The outcome of this study was the incidence of MS problems and injuries that occurred during the study course in four body regions: ankle, knee, back, and shoulder. Each answer in the OSTRC corresponds to a score. For each question (body location), a score between 0 and 25 is given, and a theoretical score (sum) ranging from 0 to 100 is calculated for four body regions. Reported scores of >39 were classified as the occurrence of MS injury (MSI; for the purpose of multinomial regression, they were later numerically scored as “2”). The presence of a MS problem (MSP) was considered if the participant scored anything higher than the lowest grade on each question (scored as “1” in regression calculation). Finally, if the participant reported a score of “zero” for all questions, the absence of any problem/injury was recorded (scored as “0” later in the regression calculation).

### 2.3. Data Analysis

In the first phase of statistical analysis, all variables were checked for the normality of distribution by the Kolmogorov–Smirnov test. Descriptive statistics calculated for variables found to be normally distributed included the means and standard deviations; otherwise, frequencies (F) and percentages (%) are reported. Injury rates are reported as the total number of injuries per studied period and the number of injuries relative to hours of exposure (dance hours; with 95% CI for Poisson rates). For these data, and irrespective of the OSTRC specific graduation, in the following text, all scores higher than minimum (zero) are collectively considered as an “injury”, if not specified otherwise.

The differences in studied categorical variables were evaluated by the chi-square test. The analysis of variance (ANOVA) was calculated in order to identify differences between/among groups, for parametric/normally distributed variables, with additional calculation of the Welch’s *p* due to unequal sample size of groups when comparison among dance styles was done.

The associations between studied predictors and outcomes (MS problem/injury) were evaluated by a univariate multinomial regression calculation using multinomial criteria based on the categorized OSTRC scale (0 = absence of MS problem/injury, 1 = MS problem, 2 = MS injury), with the absence of a problem/injury being the referent value in the multinomial regression calculation. Authors were of the opinion that usage of the multinomial regression will allow clear identification of the factors associated with MS problem and MS injury, especially knowing the differences in subjective perception of pain as an indicator of MS problem/injury. The odds ratio (OR) with the corresponding 95% confidence interval (95% CI) was reported. Multinomial regression calculations included nonadjusted regression correlations and correlations adjusted for gender, age, and dance style (Model 1). Statistica ver. 13.5 (Tibco Inc., Palo Alto, CA, USA) was used for all analyses, and a significance level of *p* < 0.05 was applied.

## 3. Results

During the study period (November/December 2018–March/April 2019), 59 dancers (47%) reported no MS injury/problem, 43% (54 dancers) reported an MS problem, and 10% (13 dancers) reported an MS injury (Figure 2). When compared across dance styles, no significant differences were obtained (Chi square = 1.51, *p* = 0.84). Altogether, 67 dancers (53%) reported at least one injury/problem, a prevalence of 39%. Multiple injuries/problems were reported by 10% of dancers. Over approximately 7050 h of dance and 91 injuries/problems occurred in total (95% CI: 73–111).

The lowest prevalence of injury occurred in females involved in S/L, with 71% of dancers experiencing no problem/injury over the study course. On the other hand, males involved in urban dances experienced highest rates of injury (18%) with additional 45% who experienced some kind of MS problem over the course of the study. The most evident difference between males and females was evidenced for standard/Latin dances, where males reported more MS problems/injury than females (71% females and 29% males reported no MS problem/injury) (Figure 3).

On average, each dancer suffered from 0.72 injuries over the study period (95% CI: 0.28–1.41) with similar rates of occurrence for both genders (0.67 (95% CI: 0.51–0.85) and 0.78 (95% CI: 0.61–0.97) in females and males, respectively), with no significant difference between genders (MW = 1.22, *p* = 0.21), a rate of 2.88 injuries per dancer per year.

Differences in the studied predictors among dance styles are shown in Table 1. The S/L dancers had been involved in dance for the shortest amount of time, but they participated in more training hours than the RNR and urban dancers did. A significant difference in BF% between individuals involved in different dance styles was shown, with SL dancers being the leanest. Differences in SBT were shown for 10 of 16 variables, and in all cases urban dancers had the lowest performance, with no significant differences between RNR and S/L dancers.

For non-adjusted regression, the higher odds for the occurrence of MS injury were shown for older dancers (OR = 1.51, 95% CI: 1.11–2.04) and for those who had had a longer career in dance (OR = 1.31, 95% CI: 1.04–1.67). Dancers with more experience in dance were at a greater risk for reporting an MS problem (OR = 1.17, 95% CI 1.04–1.32) or MS injury (OR = 1.17, 95% CI: 1.01–1.38). Gender, dance style, and anthropometric/body build indices did not significantly influence the occurrence of MS problem/injury. Several SEBT variables were correlated with MS injury/problem, with lower odds for MS injury/problem occurring in dancers who achieved better SEBT normalized scores. Namely, significant influences of dynamic balance on the occurrence of an MS problem/injury were recorded for R3 (OR (95% CI); MS injury: OR = 0.95 (0.91–0.99)), R4 (MS problem: 0.98 (0.96–0.99); MS injury: 0.97 (0.95–0.99)), R5 (MS problem: 0.98 (0.96–0.99); MS injury: 0.97 (0.95–0.99)), R6 (MS problem: 0.98 (0.96–0.99)), R7 (MS problem: 0.98 (0.96–0.99)), L3 (MS problem: 0.98 (0.95–0.99)), L5 (MS problem: 0.98 (0.97–0.99); MS injury: 0.98 (0.95–0.99)), L6 (MS problem: 0.98 (0.96–0.99); MS injury: 0.97 (0.95–0.99)), L7 (MS problem: 0.97 (0.95–0.99)), and L8 (MS problem: 0.96 (0.92–0.99)) (Table 2).

In order to statistically control the possible influences of age, gender, and dancing experience, multinomial regressions were calculated between SEBT measures, including age, gender, and dance style as confounding factors (i.e., age was a stronger predictor of an MS problem/injury than experience in dance (please see previous results), while age and dance experience were shown to be naturally correlated). Generally, even for these calculations, a similar protective influence of dynamic balance on MS problem/injury occurrence was shown, although not all variables found to be significantly correlated with the occurrence of an MS problem/injury in the “crude” model were significantly associated with the outcome in the regression model, which controlled for confounding factors (Model 1). Specifically, Model 1 showed that there was a lower likelihood of an MS problem/injury occurring for dancers who achieved better scores for R3 (OR (95% CI); MS injury: 0.96 (0.92–0.99)), L3 (MS problem: 0.98 (0.95–0.99)), L6 (MS problem: 0.98 (0.96–0.99); MS injury: 0.97 (0.95–0.99)), and L8 (MS problem: 0.97 (0.93–0.99)) (Table 2).

## 4. Discussion

With regard to the study’s aims, there are several important findings. First, the results provided no evidence of gender being a significant factor of influence on injury status. Next, a higher risk for MS injury/problem was found for older and more experienced adolescent dancers. Finally, dynamic balance was an important protective factor against MS problems/injury, irrespective of a participant’s age. Thus, the results partially support our initial study hypothesis.

### 4.1. Gender and Injury Occurrence

The injury rate of 2.8 injuries per year per dancer (280%) is somewhat lower than that presented in a recent study on Slovenian dancers, where the authors reported an injury rate of 310% [17]. However, this is not surprising since we studied younger dancers, and the risk of injury increases with age and exposure to dance [22]. Actually, the association between age and injury occurrence was confirmed in our study and will be discussed later.

We found no significant influence of gender on injury risk. This is in certain disagreement with previous reports that mostly identified a higher risk for injury among female dancers. For example, an international study revealed that females had a significantly higher median injury incidence than males and confirmed that there were gender differences regarding reported traumatic injuries, with a higher incidence of traumatic injuries occurring in females (74.6%) than males (46.7%) [27]. Supportively, another study showed that female dancers have a higher injury risk than their male counterparts [26]. Collectively, this was explained as being due to (i) the knee being the most commonly injured location, and (ii) the larger Q-angle in females that potentially translates into a greater force of the quadriceps being applied to the patella, as well as a greater likelihood for mal-tracking [26]

However, to the best of our knowledge, this is one of the first studies which examined gender as a possible factor of injury exclusively in adolescent dancers. This probably explains the discrepancy between our findings and those of previous studies where female dancers were reported to have a greater injury risk than their male peers [26,27]. We must highlight the fact that those studies involved older adolescents and adult dancers. More comparable to our study and therefore our findings is a Brazilian study of pre-professional male and female dancers (17 ± 4.44 years of age), where the authors noted no significant effect of gender on injury incidence [36]. Collectively, it seems that gender differences in injury prevalence (e.g., higher injury rate in females) are more apparent in adult than in youth dancers. However, we must note that our analysis was performed on the sample as a whole (i.e., regardless of dance style); therefore, dance-style-specific analyses are necessary. This is particularly evident if we take a closer look on gender differences in some dance styles. Specifically, in our study males involved in standard/Latin dances were more injured than their female partners. It almost certainly points to specific mechanisms of MS problem/injury for those athletes, and it deserves more attention in future investigations.

### 4.2. Age and Dance Factors as Predictors of Injury

It has already been reported that the age of a dancer can correlate with injury occurrence and risk of injury [19]. However, previous studies where injury risk was shown to increase with age regularly observed dancers with a greater age span and/or included adult dancers [37]. Meanwhile, the results of our study identified age as an important risk factor for injury occurrence, even among adolescent dancers. Our results also clearly point that an association between age and injury risk should actually be contextualized through the positive association of experience in dance (exposure to dance) and injury risk (e.g., higher injury risk occurs in dancers who have been exposed to dance for a longer period of time). Due to differences in the characteristics of samples (i.e., adolescent dancers in our study and adult dancers in previous studies), a comparison is not straightforward. Still, we can make some assumptions based on previous research findings.

Mechanical overload and excessive use, which increase with age and career length, are the most commonly reported mechanisms of injury [38]. Consequently, it is logical that more mature and experienced dancers have had greater exposure to repetitive movement patterns and are therefore at a higher risk of overload and consequent injury. In addition, greater involvement in (any) sport implies (i) a greater intensity of training and (ii) increased weekly training exposure [24,37,39]. Since injury occurs when the forces are applied to body tissues (i.e., bones, muscles) exceed the capacity of the tissue to tolerate the applied forces, increased intensity and volume of training are natural risk factors for injury occurrence [28].

Furthermore, authors investigating the prevalence and risk of MS injuries among professional dancers pointed out that with greater exposure to repetitive movement structures, functional anomalies begin to appear in dancers, leading to the adaptation of dance technique, resulting in worsened force transmission and the occurrence of microtrauma [40]. In support of this, the number of injuries per dancer is higher in professional dancers than in recreational dancers, indicating that greater dance exposure and higher technical demands correspond to an increase in risk for injury occurrence [41,42]. Currently, we are not able to identify which of the discussed factors related to age and experience in dance influence injury occurrence to the greatest extent, but some explanations are offered in the subsequent text, where we discuss the influence of balance capacity on injury occurrence.

In this study we observed different dance styles, and analyses indicated no significant influence of dance style on injury occurrence. This study is one of the first ones where injury occurrence was compared across different dance disciplines; therefore, we are not able to compare our results with those previously reported. However, from the authors’ perspective, it is possible that injury occurrence does not vary across dance styles. On the other hand, it is almost certain that dancers involved in different dance styles suffer from different types of injury and/or injured different body locations [16,21,43], which should be explored more in detail in the future.

### 4.3. Balance and Injury Occurrence

Balance is considered to be an important contributing factor to injury occurrence in sport, although not all studies have confirmed the predictive value of balance capacity on injury occurrence in athletes [28,33,44,45]. For example, balance status, as measured by the Y-balance test (a simplified version of the Star Excursion test considered in this study), was found to be predictive of injury occurrence in high school basketball players [46]. Performance in the Y-balance test was also shown to be a risk factor for injury occurrence in division I athletes [47]. Interestingly, although theoretically, balance may be associated with injury occurrence in dance, and studies have rarely examined this issue. Most probably, the issue of reliability of balance testing and the fact that balance testing is relatively time consuming (in comparison to other tests of conditioning capacities) have resulted in little empirical evidence about any association between balance and injury in dance.

To the best of our knowledge, only one very recent study has confirmed the importance of balance status in the prediction of injury occurrence in dance [17]. In brief, the study, which involved 129 competitive hip hop dancers (17.95 ± 4.15 years of age), examined predictors of injury, and a higher injury risk was shown among dancers who attained poorer scores on the explicit SEBT variables, irrespective of previous injury status. However, in the cited study, there was a certain possibility that the variation in participants’ age may have influenced both balance and injury status. Therefore, our results where balance was found to be a significant predictor of injury occurrence in adolescent dancers are novel, to some extent.

Indeed, the correlation between balance and the occurrence of an MS problem/injury in adolescent dancers is one of the important findings in this study. In brief, dancers who were shown to have better dynamic balance on the SEBT test were less likely to experience a lower extremity injury. In presenting this mechanism for the balance–injury relationship, it is important to highlight that balance is actually the ability to achieve a state of equilibrium by maintaining the body’s center of gravity over its base of support [28]. At the same time, injury results when the load applied to a structure (i.e., tissue) exceeds the capacity of the structure to sustain the load. Consequently, there are two mechanisms that can reduce the risk of injury: (i) increasing the ability of the structure to sustain the load (i.e., by strengthening the structure), and (ii) reducing the load applied to the structure [28]. Our results actually support the later mechanisms (i.e., a better balance capacity reduces the load applied to a dancer’s body structures).

Superior balance indicates better joint stability and accentuates superior neuromuscular mechanisms responsible for the co-contraction of agonists and antagonists. It actually means that dancers with better balance are more capable of achieving equilibrium and maintaining their center of gravity over the base of support [28]. In most dance forms, the base of support is the dancer’s foot, which naturally explains the here-established relationship between better balance and lower injury risk. However, it is crucial to note that the importance of balance in injury prevention seems to overcome even the previously discussed negative influence of age on injury risk in adolescent dancers. Specifically, the logistic regression analysis with balance and age as potential predictors only showed balance measured by the SEBT as a significant predictor of injury risk. Therefore, it appears that improvement in balance can decrease the risk of injury, even in those adolescent dancers whose careers last longer (i.e., older and/or more experienced dancers).

From our perspective, the explanation for our findings concerns the characteristics of dance training and competition. Namely, in all dance styles and forms, dance routines and choreography become more demanding (i.e., stressful) as the age/experience of a dancer increases [43]. The application of increased acute stress to the locomotor system (i.e., due to higher and more frequent jumps, repeated high-intensity efforts), together with more complex choreography increases the overall physiological demands of dancing, altogether resulting in a higher risk of injury occurrence. At the same time, having a superior balance capacity decreases the amount of stress applied to the body, irrespective of all specified risk factors, which occur as a result of higher dancing demands and, consequently, higher forces being applied to body structures (i.e., bones, muscles, tendons). Based on our results, the increased physiological demands of dance training and competition, known to be regular consequences of advanced dance experience, should be considered as factors with less influence on injury risk than inferior dynamic balance.

Additional (supplementary) balance training aimed at the prevention and rehabilitation of MS injuries, as well as improving sport performance, have become increasingly popular in sports [48,49]. Specific equipment has constantly been developed and is used in this type of training (i.e., balance balls, semicircular platforms, slack line, different types of balance platforms, rotator discs). Collectively, various exercise modalities are confirmed to be effective in improvement of dynamic balance in youth, even those involved in competitive sports [50,51]. Literature suggests that positive effects may be expected from relatively short training sessions (4–15 min of workout per session), performed twice per week, while largest effects may be expected after 12 weeks of training, resulting in 24–36 training sessions in total [51]. Therefore, selected balance exercises can be elegantly included as a part of warm-up session several times per week, assuring the low-cost, and effective stimuli are aimed at improving the dynamic balance, even in youth dancers.

### 4.4. Limitations and Strengths

There are several limitations of this study. First, we must highlight the unequal number of dancers participating in the different styles. Therefore, although the analyses performed did not indicate a significant influence of dance style on injury occurrence, it is still possible that the results are not equally generalizable to all studied dance styles. Second, there is a certain possibility that balance status was actually altered by some indices not examined in this study. Next, we certainly did not observe all factors potentially related to injury occurrence, such as biological age, motor competence, conditioning status, etc. Therefore, in future studies, special attention should be paid to other indices of a dancer’s status and their influences on injury occurrence. In addition, the outcome was measured using the OSTRC, and this measurement tool examines injury occurrence at four body sites. On the other hand, it is possible that dancers suffered from specific injuries at other locations (i.e., wrist, neck).

To the best of our knowledge, this is one of the first studies to examine injury occurrence and factors associated with injury occurrence exclusively in adolescent dancers involved in the most popular dance styles. The study used a low-cost and applicable measurement tool for the evaluation of balance status. Since we found a significant influence of balance, as measured by the SEBT, our results are applicable for various circumstances. Next, all participants were measured in the same facility by experienced evaluators. Finally, the prospective nature of the study and the consequent lack of recall bias are important strengths of the investigation.

## 5. Conclusions

This study provides evidence of the negative influences of age and experience on injury occurrence among adolescent dancers. This is a logical consequence of greater sport demands (as a result of increased complexity in dancing routines and choreography) and a higher volume of training. Therefore, even in youth dancers, in order to prevent injury, special attention should be placed on more experienced dancers.

Higher risk for injury was found in dancers with lower results in the SEBT. While this testing protocol is simple, reliable, cheap, and applicable in different circumstances, we suggest that regular screening of dynamic balance in dancers should occur. This will allow the identification of dancers with a potential risk for injury occurrence.

Analyses performed in this investigation showed that dynamic balance is a more important predictor of injury occurrence than age (experience in dance). Therefore, it is expected that improvement in balance could diminish the risk of injury in dancers, irrespective of their age/experience. In order to achieve a stable position during choreographed movements, dancers must continually improve their balance and thus reduce their injury risk. Collectively, we suggest that specific interventions/training aimed at the improvement of dynamic balance are important components of injury prevention for adolescent dancers.

## Figures and Tables

**Figure 1 children-07-00297-f001:**
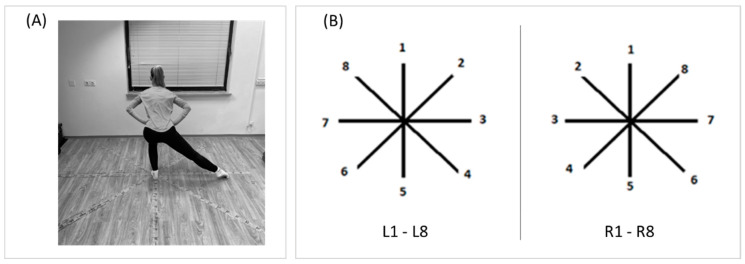
Execution of the Star Excursion Balance Test (**A**), and scoring while standing on the left leg (L1–L8) and while standing on the right leg (R1–R8) (**B**).

**Figure 2 children-07-00297-f002:**
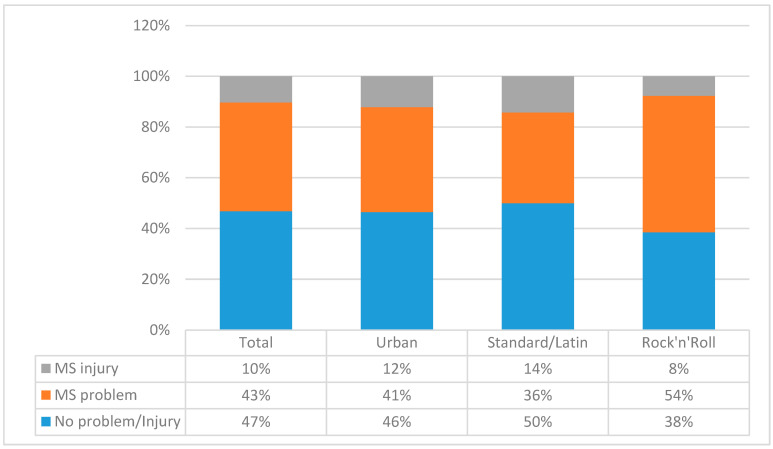
Prevalence of the musculoskeletal (MS) problems and injures in adolescent dancers.

**Figure 3 children-07-00297-f003:**
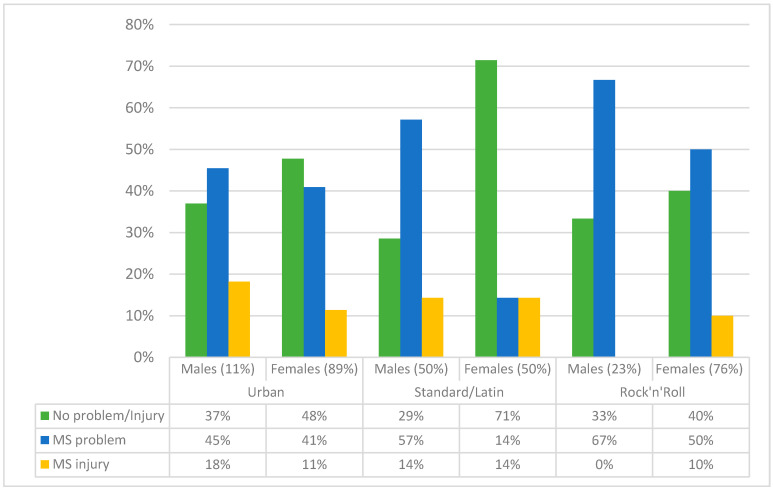
Prevalence of the musculoskeletal (MS) problems and injuries in adolescent dancers according to gender for studied dance styles.

**Table 1 children-07-00297-t001:** Descriptive statistics (presented as means ± standard deviations) and differences between dance styles in studied variables (ANOVA with additional Welch’s *p* calculation).

	Urban Dances	Standard/Latin Dances	Rock and Roll	ANOVA		Welch’s *p*
	(*n* = 99)	(*n* = 14)	(*n* = 13)	F Test	*p*	
Age (years)	15.8 ± 1.4	15.25 ± 1.96	15.57 ± 1.66	1.42	0.241	0.031
Started with dance (years)	7.77 ± 2.8	9.3 ± 2.7	7.52 ± 3.25	3.08	0.052	0.294
Involvement in dance (years)	8.11 ± 2.65	6 ± 2.65	8.29 ± 2.72	6.47	0.001	0.001
Trainings per week (f)	3.42 ± 0.87	5.57 ± 1.53	5.1 ± 0.89	63.09	0.001	0.001
Hours of training per week (h)	5.41 ± 1.77	17.91 ± 6.13	11.14 ± 2.39	205.47	0.001	0.001
Body height (cm)	165.85 ± 6.33	167.23 ± 9.97	166.91 ± 10.93	0.44	0.651	0.832
Body mass (kg)	58.92 ± 7.01	55 ± 11.38	58.68 ± 13.33	2.01	0.143	0.082
Body fat (%)	21.89 ± 6.51	12.09 ± 5.26	15.51 ± 5.63	29.52	0.001	0.001
Lean body mass (%)	25.34 ± 3.39	26.91 ± 6.65	27.54 ± 6.95	2.97	0.052	0.567
Body mass index (kg/m^2^)	21.43 ± 2.44	19.45 ± 2.1	20.83 ± 2.58	6.71	0.001	0.001
R1 (SEBT result/leg length)	83.86 ± 9.61	85.35 ± 5.88	83.29 ± 6.6	0.32	0.721	0.022
R2 (SEBT result/leg length)	89.25 ± 12.36	95.8 ± 12.79	92.21 ± 6.38	3.00	0.052	0.031
R3 (SEBT result/leg length)	100.23 ± 14.08	124.77 ± 23.14	111.32 ± 12.83	24.83	0.001	0.001
R4 (SEBT result/leg length)	105.72 ± 16.31	128.73 ± 20.3	130.17 ± 14.08	31.70	0.001	0.001
R5 (SEBT result/leg length)	106.1 ± 16.82	132.11 ± 19.63	134.72 ± 20.57	37.03	0.001	0.001
R6 (SEBT result/leg length)	100.07 ± 16.68	125.27 ± 22.6	127.35 ± 23.22	31.42	0.001	0.001
R7 (SEBT result/leg length)	89.04 ± 15.75	114.04 ± 22.76	106.71 ± 16.45	25.78	0.001	0.001
R8 (SEBT result/leg length)	74.84 ± 10.02	75.6 ± 11.36	75.23 ± 5.37	0.06	0.94	0.323
L1 (SEBT result/leg length)	83.52 ± 9.96	81.72 ± 5.42	81.25 ± 7.67	0.75	0.471	0.541
L2 (SEBT result/leg length)	89.82 ± 10.04	90.99 ± 8.3	90.85 ± 7.55	0.20	0.823	0.289
L3 (SEBT result/leg length)	99.2 ± 14.65	116.9 ± 20.78	116.79 ± 13.3	19.97	0.001	0.001
L4 (SEBT result/leg length)	107.38 ± 16.26	129.77 ± 20.22	130.52 ± 14.23	28.69	0.001	0.001
L5 (SEBT result/leg length)	106.71 ± 17.92	132.19 ± 21.65	135.68 ± 17.47	33.52	0.001	0.001
L6 (SEBT result/leg length)	100.57 ± 17.57	127.77 ± 23.2	125.65 ± 19.59	30.13	0.001	0.001
L7 (SEBT result/leg length)	90.12 ± 14.92	110.84 ± 23.65	107.39 ± 17.89	20.16	0.001	0.001
L8 (SEBT result/leg length)	74.48 ± 11.99	77.95 ± 10.23	77.13 ± 13.53	1.02	0.364	0.046

R1–R8—normalized result on Star Excursion Balance Test (SEBT) while standing on the right leg in eight directions; L1–L8—normalized result on Star Excursion Balance Test (SEBT) while standing on the left leg.

**Table 2 children-07-00297-t002:** Correlates of musculoskeletal problems and injury in adolescent dancers; results are given as OR (95% CI).

	Model 0 (Nonadjusted)		Model 1 (Adjusted for Dance Style, Age, and Gender)	
	MS Problem	MS Injury	MS Problem	MS Injury
Male gender	1.57 (0.58–4.21)	1.05 (0.39–2.76)		
Age (years)	1.09 (0.89–1.33)	1.51 (1.11–2.04)		
Involvement in dance (years)	1.05 (0.91–1.20)	1.31 (1.04–1.66)	1.01 (0.88–1.23)	1.11 (0.91–1.51)
Training per week (hours)	1.05 (0.93–1.25)	1.00 (0.87–1.17)	1.03 (0.91–1.27)	1.01 (0.85–1.20)
Body height (cm)	1.00 (0.97–1.05)	0.99 (0.94–1.05)	1.00 (0.96–1.07)	1.00 (0.93–1.06)
Body mass (kg)	1.02 (0.98–1.06)	1.02 (0.97–1.07)	1.01 (0.96–1.08)	1.01 (0.96–1.08)
Body fat (%)	1.00 (0.97–1.06)	1.05 (0.97–1.15)	1.01 (0.98–1.08)	1.06 (0.98–1.17)
Lean body mass (kg)	1.01 (0.95–1.08)	0.99 (0.92–1.08)	1.02 (0.94–1.09)	0.99 (0.91–1.10)
Body mass index (kg/m^2^)	0.97 (0.83–1.14)	0.98 (0.81–1.21)	0.95 (0.81–1.15)	0.95 (0.80–1.23)
R1 (SEBT result/leg length)	0.97 (0.92–1.03)	0.93 (0.86–1.00)	0.98 (0.90–1.05)	0.93 (0.86–1.00)
R2 (SEBT result/leg length)	0.99 (0.95–1.03)	0.94 (0.89–1.01)	1.00 (0.92–1.05)	0.96 (0.87–1.04)
R3 (SEBT result/leg length)	0.99 (0.97–1.01)	0.95 (0.91–0.99)	0.98 (0.94–1.01)	0.96 (0.92–0.99)
R4 (SEBT result/leg length)	0.98 (0.96–0.99)	0.97 (0.95–0.99)	0.99 (0.94–1.03)	0.98 (0.92–1.03)
R5 (SEBT result/leg length)	0.98 (0.96–0.99)	0.97 (0.95–0.99)	0.99 (0.94–1.02)	0.98 (0.93–1.03)
R6 (SEBT result/leg length)	0.98 (0.97–0.99)	0.98 (0.96–1.01)	0.99 (0.95–1.04)	0.98 (0.94–1.02)
R7 (SEBT result/leg length)	0.98 (0.96–0.99)	0.99 (0.96–1.01)	0.97 (0.94–1.01)	0.98 (0.95–1.02)
R8 (SEBT result/leg length)	0.96 (0.93–1.01)	1.01 (0.96–1.06)	0.98 (0.92–1.04)	1.00 (0.95–1.06)
L1 (SEBT result/leg length)	0.98 (0.93–1.02)	0.98 (0.93–1.03)	0.99 (0.94–1.05)	1.00 (0.94–1.04)
L2 (SEBT result/leg length)	0.99 (0.95–1.03)	0.99 (0.94–1.04)	0.99 (0.95–1.04)	0.99 (0.94–1.05)
L3 (SEBT result/leg length)	0.98 (0.96–0.99)	0.98 (0.96–1.01)	0.98 (0.95–0.99)	0.98 (0.95–1.02)
L4 (SEBT result/leg length)	0.98 (0.97–1.01)	0.98 (0.96–1.01)	0.99 (0.97–1.02)	1.00 (0.97–1.03)
L5 (SEBT result/leg length)	0.98 (0.97–0.99)	0.98 (0.95–0.99)	0.99 (0.97–1.02)	1.00 (0.95–1.05)
L6 (SEBT result/leg length)	0.98 (0.96–0.99)	0.97 (0.95–0.99)	0.98 (0.96–0.99)	0.97 (0.95–0.99)
L7 (SEBT result/leg length)	0.97 (0.95–0.99)	0.98 (0.95–1.01)	0.98 (0.95–1.01)	0.99 (0.95–1.03)
L8 (SEBT result/leg length)	0.96 (0.92–0.99)	0.97 (0.93–1.01)	0.97 (0.93–0.99)	0.98 (0.92–1.03)

R1–R8—normalized result on Star Excursion Balance Test (SEBT) while standing on the right leg in eight directions; L1–L8—normalized result on Star Excursion Balance Test (SEBT) while standing on the left leg.

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
