# Peer review of "Predicting Injury Status in Adolescent Dancers Involved in Different Dance Styles: A Prospective Study"

_children, 2020, doi:10.3390/children7120297_

Round 1

Reviewer 1 Report

I included the Word file here. 

Author Response

REVIEWER 1

Title: Predicting injury status in adolescent dancers involved in difference dance styles: A prospective study

SPECIFIC COMMENTS TO EDITOR AND/OR AUTHOR(S)

To the Author(s):

I appreciate this manuscript's topic and recognize meaningful information to value a unique group: Dancers. The premise behind this manuscript is robust and would be of interest to the readers of the journal. However, there are some severe issues that the authors need to properly examine/address for this work to make an impact. In the following, I will discuss the specific problems that should be addressed by a specific line-by-line critique.

RESPONSE: Thank you for recognizing the potential of our work. Also, we are particularly grateful to your constructive and elaborated comments and suggestions. We tried to follow it strictly and amended the manuscript. Please see below for specific responses. Staying at your disposal.

Overall, there are severe writing issues in the consistency of using the abbreviation. For example, the authors provided the abbreviation, physical activity (PA), in line 34, but in line 48, full words (physical activity) were mentioned again. Additionally, in line 50, musculoskeletal (MS) injury was stated, but in line 78, the full word, musculoskeletal, was shown again. Throughout the manuscript, there are many severe inconsistencies using abbreviations in the terms, musculoskeletal problems and injury. Please, review the whole manuscript and adjust the issues.

RESPONSE: In the revised  version of the manuscript we paid special attention on consistency of abbreviations. Thank you for noticing this issue.

Specifically;

41-42: I appreciate that the authors used several articles that have been published in Children. I strongly suggest including one more article, which is related to the PA intervention programs for young children that has been published this year in the Children:

Lee, J., Zhang, T., Chu, T. L., & Gu, X. (2020). Effects of a need-supportive motor skill intervention on children’s motor skill competence and physical activity. Children, 7(3), 21. https://doi.org/10.3390/children7030021

RESPONSE: Thank you for your suggestion. The mentioned article is included (please see first paragraph of the Introduction).

104-106: It is not clear to provide a reasonable rationale about the statistical power, specifically HOW the authors received the results.  I would like to see what program the author used to calculate the power and what information regarding the data analysis did the author input instead of describing the previous study's method.

RESPONSE: Indeed, the calculation of the sample size was not properly described. Text now reads: “Based on (i) an previously evidenced injury occurrence of 10% [1], (ii) a population sample of 621registered adolescent dancers in Slovenia for studied year, (iii) margin of error of 0.05, and (iii) confidence level of 0.95, the required sample size for this investigation was calculated to be 114 participants (calculated by Statistica, Tibco Inc.)” – Please see highlighted text in Participants subsection (note that it can be also calculated by other tools, such as https://goodcalculators.com/sample-size-calculator/)

107: Study design and variables. This section needs to be reorganized. Please, provide subheadings on each measurement (variable). For example, 2.2.1. Sociodemographic; 2.2.2. Anthropometrics…….

RESPONSE: Reorganized as suggested. Please see Methods section. Thank you.

109: Please, change from December 2019/January 2020 to December 2019-January 2020. Ĺ˝

RESPONSE: Amended as suggested, but please see next comment and response. Thank you.

111: The authors mentioned the follow-up testing (over a period of 3 months after baseline), which means the testing was conducted March-April 2020. How did you conduct the research during the pandemic? It would be necessary to provide more specific information regarding the follow-up testing. I also cannot see the results regarding the following test results on the Results section.

RESPONSE: Thank you for this notation. It was our huge mistake. Our initial screening was done in late 2018, and follow-up during March-April 2019. It is now corrected.

151-152: Although the SEBT test has shown the great inter- and intra-rater reliability in the previous studies, this study did not provide the inter-rater reliability, which might occur the examiner bias in this study. The authors need to include this issue in the limitation section.

RESPONSE: Actually, we didn’t have the problem of inter-rater reliability because all dancers were evaluated by same examiner. It is now clearly noted in the text where details on SEBT measurement were provided. Text reads “All dancers were evaluated by same examiner in the same facility (Faculty of Sport, Ljubljana).”

163: Please, check it: left leg (L1–L8)

RESPONSE: Corrected, thank you.

165: Figure 1: I can see the different dash style on the image (B) between L1 – L8 and R1 – R8. Please, keep consistency.

RESPONSE: Unified. Thank you.

168: Statics -> Data analysis

RESPONSE: Amended as suggested.

172: Injury rates are reported as the total number of injuries per studied period and the number of injuries relative to hours of exposure (dance hours; with 95%CI for Poisson rates). It would be important to provide the study period as numbers right next to “studied period” For example, studied period (December 2020–January 2021).

RESPONSE: Thank you. The text is amended accordingly. It reads: “During the study period (November/December 2018 – March/April 2019), 59 dancers (47%) reported no MS injury/problem, 43% (54 dancers) reported an MS problem, etc.” (please see first part of the Results section).

177-186: It would be significant to provide the information “Why” and “What” made you decide to use those statistical analysis? Especially, I would like to see clear information “Why” authors used multinomial regression on this study.

RESPONSE: Indeed, the specific reason for using multinomial regression was not specified. It is now explained, and text reads: “Authors were of the opinion that usage of the multinomial regression will allow clear identification of the factors associated with MS problem and MS injury, especially knowing the differences in subjective perception of pain as an indicator of MS problem/injury.” (please see Data analysis subsection).

182: Please, properly use the symbols. The authors need to use equal sign: 0 = absence of MS problem/injury, 1 = MS problem, 2 = MS injury

RESPONSE: Amended accordingly.

188: p should be italic p in statistical writing manner. Please, change them throughout the manuscript.

RESPONSE: Corrected, thank you.

196: Figure 2. In the Urban group, the sum of percentages is not 100%. 

RESPONSE: As you probably guessed, it was because of “rounded numbers” (no decimals). It is corrected in the revised version. Thank you.

203: Figure 3.

1) Please, add the information about the total percentages of males and females;

RESPONSE: Added.

2) Please, add the words MS problem and MS injury on the category variables;

RESPONSE: Added

3) the sum of percentage in Urban group is not 100%.

RESPONSE: Thank you for noticing it (problem with rounded numbers again). It is corrected.

4) Female has significantly higher percentage (71%) in Standard/Latin group on No problem/injury variable. Male has significantly higher percentage (57%) in Standard/Latin group on MP problem. However, the authors did not mention any group differences (urban, standard/Latin, rock’s’ roll) on the Results and Discussion parts. The readers would be curious why there are many MS problem among males compared to females. Since this is the one of research question in this study, it is important to provide the results.

RESPONSE: Thank you for mentioning this. In the revised version of the manuscript it is now presented in Results section. Text reads: “The most evident difference between males and females was evidenced for Standard/Latin dances, where males reported more MS problems/injury than females (71% females and 29% males reported no MS problem/injury)”. Also, this is now briefly discussed. Text reads: “However, we must note that our analysis was performed on the sample as a whole (i.e., regardless of dance style), and, therefore, dance-style-specific analyses are necessary. This is particularly evident if we take a closer look on gender differences in some dance styles. Specifically, in our study males involved in Standard/Latin dances were more injured than their female partners. It almost certainly points to specific mechanisms of MS problem/injury for those athletes, and deserves more attention in future investigations.” (please see end of subsection 4.1).

 220: Table 2. There is severe issue regarding the unequal group sample size. How did the authors handle the unequal sample size? If you just ran the analysis, the results could be influenced by the unequal sampling. The authors need to check the Brown-Forsythe & Welch F tests to deal with the unequal sampling.

RESPONSE: Thank you for your suggestion. In this version of the manuscript we additionally calculated Welch p. Text in Data analysis subsection reads: “The differences in studied categorical variables were evaluated by the chi-square test. The analysis of variance (ANOVA) was calculated in order to identify differences between/among groups, for parametric/normally distributed variables, with additional calculation of the Welch p due to unequal sample size of groups when comparison among dance styles was done.” (please see Data analysis subsection) 

223: “Results of the nonadjusted multinomial regressions calculated for each observed predictor and injury occurrence are presented in Table 2.” I don’t the table present the “nonadjusted multinomial regressions” It was the descriptive statistics with ANOVA results. The authors need to mention about the ANOVA analysis on the “Data analysis” part.

RESPONSE. Thank you for noticing this mistake in referencing the Tables in the text of Results. It is corrected now However, we must not that tables of non-adjusted and adjusted logistic regressions are combined in one (now Table 2). Also, ANOVA is specified in Data analysis part.

240 and 258: Table 3 and 4. 1) The variable should either musculoskeletal problem or injury. Please, change the term on the category variable;

RESPONSE: Thank you for noticing it. It is corrected. Please see Table 2 now.

2) It would be better to combine Table 3 and 4 to increase visualization so that readers can easily see the differences.

RESPONSE: Tables 3 and 4 are combined in one as suggested. Please see Table 2 in this version of the manuscript.

244: “Multinominal logistic regressions” In the data analysis, the authors did not mention about this term. It would be important to keep consistency of using the term.

RESPONSE: Thank you. It is corrected (multinomial regression).

261: Discussion. I would like to see information about the different dance styles and how these factors influence their MS problems and injury, but the information is missing. On line 291-293, the authors noted that the specific data analysis for the dance styles were not conducted. However, the study examined the different dance styles; thus, it would be essential to measure it or clarify “why” this study did not apply for the specific analysis for the different dance styles.

RESPONSE: We must agree that originally we didn’t elaborate possible influence of “dance styles” on injury. However, this was because statistical analyses didn’t provide sufficient evidence about association of dance style with injury status (please see Results where differences among dance styles are presented in Table 1, compared by Chi square test). However, we also agree that this issue deserves attention. Therefore, in this version of the manuscript the problem is briefly discussed irrespective of “lack of influence of dance styles on injury”, and text reads: “In this study we observed different dance styles, and analyses indicated no significant influence of dance style on injury occurrence. This study is one of the first ones where injury occurrence was compared across different dance disciplines, and therefore we are not able to compare our results with those previously reported. However, from authors’ perspective, it is possible that injury occurrence doesn’t vary across dance styles. On the other hand, it is almost certain that dancers involved in different dance styles suffer from different type of injury, and/or injured different body locations [16,21,43], which should be explored more in details in future.” (please see last paragraph of the subsection 4.2 Age and dance factors as predictors of injury). Thank you.

390: Again, inconsistency of using abbreviation for the SEBT.

RESPONSE: Corrected.

401: Please, change the term, “cheap,” to “low-cost.”

RESPONSE: Changed as suggested.

407-408: It would be important to add the detailed application here. “How” can dancers improve their dynamic balances. Specific information would be helpful for understanding and providing practical information for dancers to prevent their injury. The authors may provide coaching approach.

RESPONSE: We are particularly grateful for this suggestion. In this version of the manuscript this practical issue is specifically highlighted, and text reads: “Additional (supplementary) balance training became increasingly popular in sports over the last decade, and is mostly oriented toward prevention and rehabilitation of MS injuries, as well as improving sport performance [2,3]. Also, specific equipment is developed and used in this type of training (i.e. balance balls, semicircular platforms, slack line, different types of balance platforms, rotator discs). Finally, various exercise modalities are confirmed to be effective in improvement of dynamic balance in youth, even those involved in competitive sports [4,5]. Importantly, positive effects may be expected from relatively short training sessions (4-15 min of workout per session), performed twice per week, and largest effects may be expected after 12-weeks of training, resulting in 24-36 training sessions in total [5]. Therefore, selected balance exercises can be elegantly included as a part of warm-up session several times per week, assuring the time non-consuming, low-cost, and effective stimuli aimed at improvement of this important capacity even in youth dancers.” (please see last paragraph of the subsection 4.3. Balance and injury occurrence

Staying at your disposal for any further suggestion 

Authors

Reviewer 2 Report

Congratulations on the work presented.

I present in the attached document a series of recommendations that I hope will be useful to you.

Author Response

REVIEWER 2

I have enjoyed reading your manuscript, which is interesting, rigorous, and adequate. However, I am going to make some suggestions for improvement that I think can contribute to increasing the quality of the manuscript.

RESPONSE: Thank you for recognizing the quality of our research and manuscript. Also, thank you for your comments and suggestions. We tried to follow it specifically and amended the manuscript accordingly. Please see RESPONSES in the following text.

1) Abstract

  • Avoid acronyms that are not necessary in the abstract, for example musculoskeletal problems (MSP). I think these acronyms in the body of the text would be more recommended

RESPONSE: Amended accordingly. Abbreviations are omitted in the Abstract

2) Introduction and Discussion

In lines 34 to 40 when talking about the health benefits of practicing physical activity, the following articles could be cited, their topicality and subject matter reinforces this argument:

  • Moral-García, J. E., Agraso-López, A. D., Ramos-Morcillo, A. J., Jiménez, A., & Jiménez-Eguizábal, A. (2020). The influence of physical activity, diet, weight status and substance abuse on students’ self-perceived health. International Journal of Environmental Research and Public Health, 17(4), 1387. https://doi.org/10.3390/ijerph17041387
  • Moral-García, J. E., Agraso López, A. D., Pérez Soto, J. J., Rosa Guillamón, A., Tárraga Marcos, M. L., García Cantó, E., & Tárraga López, P. J. (2019). Práctica de actividad física según adherencia a la dieta mediterránea, consumo de alcohol y motivación en adolescentes. Nutrición Hospitalaria, 36(2), 420-427. http://dx.doi.org/10.20960/nh.2181

Including information related to these jobs can be very helpful. I suggest citing the following articles.

  • Moral-García, J. E., Urchaga-Litago, J. D., Ramos-Morcillo, A. J., & Maneiro, R. (2020). Relationship of Parental Support on Healthy Habits, School Motivations and Academic Performance in Adolescents. International journal of environmental research and public health, 17(3), 882. https://doi.org/10.3390/ijerph17030882
  • Serna, C.; Martínez, I. Parental Involvement as a Protective Factor in School Adjustment among Retained and Promoted Secondary Students. Sustainability 2019, 11(24), 7080; https://doi.org/10.3390/su11247080.
  • Garcia, F.; Serra, E.; Garcia, O.F.; Martinez, I.; Cruise, E. A Third Emerging Stage for the Current Digital Society? Optimal Parenting Styles in Spain, the United States, Germany, and Brazil.  J. Environ. Res. Public Health2019, 16, 2333.

Classic jobs can also be used:

  • Spera, C. (2005). A review of the relationship among parenting practices, parenting styles, and adolescent school achievement. Educational Psychology Review, 17, 125-146.
  • Steinberg, L., Lamborn, S. D., Dornbusch, S. M., & Darling, N. (1992). Impact of parenting practices on adolescent achievement: Authoritative parenting, school involvement, and encouragement to succeed. Child Development, 63, 1266-1281.

RESPONSE: Thank you for your suggestions. In this version of the manuscript several of the previously mentioned references are included, specifically:

  • Moral-García, J. E., Urchaga-Litago, J. D., Ramos-Morcillo, A. J., & Maneiro, R. (2020). Relationship of Parental Support on Healthy Habits, School Motivations and Academic Performance in Adolescents. International journal of environmental research and public health, 17(3), 882. https://doi.org/10.3390/ijerph17030882
    • Included in the Introduction. Text reads: “The importance of participating in sufficient physical activity (PA) in childhood and adolescence and the benefits of PA on physical health, mental health, academic performance, and social well-being have been well proven [1-5].

  • Moral-García, J. E., Agraso-López, A. D., Ramos-Morcillo, A. J., Jiménez, A., & Jiménez-Eguizábal, A. (2020). The influence of physical activity, diet, weight status and substance abuse on students’ self-perceived health. International Journal of Environmental Research and Public Health, 17(4), 1387. https://doi.org/10.3390/ijerph1704138
    • Included in the Introduction. Text reads: “However, the decrease of participation in PA in adolescence is recognized as a global problem, which is additionally important since a decrease in PA may extend into adult life [6-8].”

3) Materials and Methods (line 83 to 187). Some suggestions are proposed.

  • Distribute in the following sections: Design and Participants; Instruments; Process; Data analysis.

RESPONSE: Divided as suggested.

  • Explain the process of selecting the sample of participants (in participants)

RESPONSE: Selection process in explained as you suggested. Text reads: “Participants were selected on the basis of their status in dance sport, and all participants should be regular competitors at national and international levels under the auspices of the Slovenian Dance Association (SDA) and the International Dance Organization at the study baseline.” (please see Participants subsection – highlighted text).

  • Explain the inclusion and exclusion criteria to participate in this study (in procedures).

RESPONSE: The inclusion/exclusion criteria are specified and text reads: “Dancers were qualified for inclusion if they were (i) minors (18 years or younger at the end of the study—see later for design); (ii) officially registered as a competition participant in the International Dance Organization; and (iii) participating in at least 2 dance trainings per week. Exclusion criteria included age of 18+ at the end of the study, non-regular participation in dance training (less than 2 dance trainings per week as defined by main coach), and injury/sickness during the baseline testing (see later for details on testing).” (please see Participants subsection – highlighted text).

I congratulate the authors, especially, for the development of section “2.2. Study design and variables”.

RESPONSE: Thank you

4) Results

  • They have been addressed correctly.

RESPONSE: Thank you

5) Discussion

  • I consider that the citations used are up to date, but the inclusion of the citations of the aforementioned works is recommended (in point 2. Introduction and Discussion).

RESPONSE: Thank you for your suggestions. The citations are included (please see previous comments and responses).

  • You can also try to explain the findings by trying to establish more causal relationships between the variables analyzed.

RESPONSE: In this version of the manuscript we paid attention on this issue. For example we discussed the issue of “dance experience” on “increase of the training load”, and consequently on higher injury risk among dancers (please see highlighted text in Discussion section). Also, we mentioned this issue in Limitations section, and text reads: “Second, there is a certain possibility that balance status was actually altered by some indices not examined in this study.” (please see limitations of the study – end of Discussion).

6) Conclusions

  • I think that's fine.

RESPONSE: Thank you.

7) Bibliographic references

  • I think that's fine.

RESPONSE: Thank you.

With all the humility these recommendations are collected with the intention that they can be of help to improve this work.

Congratulations on your research.

Once again, we must thank you for your support and suggestions.

Staying at your disposal

Authors

Round 2

Reviewer 1 Report

Hello, thank you for taking the time to make all of the (tedious) revisions I requested! I appreciate your hard work and recognize that this document is ready for publication.